# Interferon Signaling-Dependent Contribution of Glycolysis to Rubella Virus Infection

**DOI:** 10.3390/pathogens11050537

**Published:** 2022-05-03

**Authors:** Erik Schilling, Maria Elisabeth Wald, Juliane Schulz, Lina Emilia Werner, Claudia Claus

**Affiliations:** 1Institute of Clinical Immunology, Medical Faculty, Leipzig University, Johannisallee 30, 04103 Leipzig, Germany; erik.schilling@medizin.uni-leipzig.de; 2Institute of Virology, Faculty of Veterinary Medicine, Leipzig University, An den Tierkliniken 29, 04103 Leipzig, Germany; mw28guqu@studserv.uni-leipzig.de; 3Institute of Medical Microbiology and Virology, Medical Faculty, Leipzig University, Johannisallee 30, 04103 Leipzig, Germany; juliane.schulz@student.uni-halle.de (J.S.); lina.werner@medizin.uni-leipzig.de (L.E.W.)

**Keywords:** extracellular acidification rate, extracellular flux analysis, glycolysis, oxygen consumption rate, 2-deoxy-D-glucose

## Abstract

Interferons (IFNs) are an essential part of innate immunity and contribute to adaptive immune responses. Here, we employed a loss-of-function analysis with human A549 respiratory epithelial cells with a knockout (KO) of the type I IFN receptor (IFNAR KO), either solely or together with the receptor of type III IFN (IFNAR/IFNLR1 KO). The course of rubella virus (RuV) infection on the IFNAR KO A549 cells was comparable to the control A549. However, on the IFNAR/IFNLR1 KO A549 cells, both genome replication and the synthesis of viral proteins were significantly enhanced. The generation of IFN β during RuV infection was influenced by type III IFN signaling. In contrast to IFNAR KO A549, extracellular IFN β was not detected on IFNAR/IFNLR1 KO A549. The bioenergetic profile of RuV-infected IFNAR/IFNLR1 KO A549 cells generated by extracellular flux analysis revealed a significant increase in glycolysis, whereas mitochondrial respiration was comparable between all three cell types. Moreover, the application of the glucose analogue 2-deoxy-D-glucose (2-DG) significantly increased viral protein synthesis in control A549 cells, while no effect was noted on IFNAR/IFNLR KO A549. In conclusion, we identified a positive signaling circuit of type III IFN signaling on the generation of IFN β during RuV infection and an IFN signaling-dependent contribution of glycolysis to RuV infection. This study on epithelial A549 cells emphasizes the interaction between glycolysis and antiviral IFN signaling and notably, the antiviral activity of type III IFNs against RuV infection, especially in the absence of both type I and III IFN signaling, the RuV replication cycle was enhanced.

## 1. Introduction

The involvement of cellular metabolism during virus infections can be discussed from two perspectives. First, cellular metabolism contributes to the host cell’s antiviral response mechanisms. Second, virus infection-induced metabolic alterations up to reprogramming of host cell metabolism support the increased need for energy and metabolites during virus infections. The multiple points of interaction between interferon (IFN) response, virus infection, and metabolism have been discussed in several reviews including recent reviews by Thaker et al. on viral manipulation of cellular metabolism [1] and by Choi et al. and Sachez-Garcia et al. on the role of the tricarboxylic acid (TCA) cycle as a central metabolic pathway [2,3]. TCA cycle intermediates can support virus infections (e.g., through the influx of citrate into fatty acid biosynthesis), but also the antiviral response (e.g., through itaconate that is derived from cis-aconitate). This also highlights the relevance of cellular metabolism for the study of antiviral treatment options such as the application of metabolic intermediates or inhibitors for metabolic pathways [4]. 

As an important first line antiviral defense mechanism, the innate immune response involves the secretion of several cytokines and chemokines including IFNs. The antiviral activity of IFNs includes an antiproliferative activity. This is also part of their antitumor activity, which was, for example, assigned to type I IFN β against ex vivo peripheral blood mononuclear cells of patients with adult T-cell leukemia/lymphoma [5]. Our group recently demonstrated that the influence of rubella virus (RuV) infection on the metabolism of human macrophages can in most parts be attributed to IFN β formed during RuV infection. Exogenous IFN β induced a loss of the glycolytic reserve in human macrophages as was noted for RuV [6]. Olson et al. reported similar results through their observation on the contribution of IFN β to the reduction in the glycolytic reserve in murine bone marrow-derived macrophages during infection with *Mycobacterium tuberculosis* [7].

In the case of the positive-sense single-stranded (+ss)RNA virus RuV, as a member of the family *Matonaviridae*, the innate immune response is not well studied. In vivo patient data of a homozygous mutation in the IFN α/β receptor (IFNAR2) support the relevance of the IFN response as an antiviral countermeasure against RuV. Vaccine-derived RuV was identified in the brain biopsy of a young patient with fatal encephalitis, whereas an altered susceptibility to other viruses prior to MMR (measles, mumps, and rubella) vaccination was not described [8]. Thus, the study of IFN response during RuV infection has also implications for RuV pathogenesis such as IFNAR2 and STAT2 deficiencies in humans, which especially for MMR vaccine viruses are associated with an increased viral susceptibility [9]. In cell culture pre-infection application of IFN β reduces viral protein synthesis and the number of extracellular virus particles significantly, whereas a comparable, but slightly reduced and RuV strain-dependent effect of IFN λ was noted [10].

Here, we hypothesized that the metabolic alterations induced by RuV on different human and animal cell lines [11] are influenced by the associated IFN response. RuV-induced alterations in metabolic pathways including glycolysis were especially noted on Vero cells [11]. Vero cells lack the ability to synthesize type I IFNs, but respond to exogenous type I IFNs as the components of the signaling pathway are maintained [12,13]. The contribution of IFN signaling to virus infection-associated alterations in glycolysis was also shown for Usutu virus (USUV) infection on Vero cells, as after addition of exogenous IFN β to USUV-infected Vero, the observed increase in glycolysis was lost [14]. Vero E6 cells were also employed in a recent study on SARS-CoV-2, which revealed an increase in glucose metabolism during infection and a reduction in virus replication and infectivity of virus particles after the application of the glycolysis inhibitor 2-deoxy-D-glucose (2-DG) [15]. We studied the influence of IFN-associated signaling pathways on cellular metabolism during RuV infection of A549 cells with a knock-out (KO) of the type I IFN receptor either solely (IFNAR KO) or together with type III IFN receptor (IFNAR/IFNLR1 KO). On epithelial A549 cells, a robust type I and III IFN response was induced by RuV together with an increase in mitochondrial respiration [10,11]. We found that on IFNAR/IFNLR1 KO A549 cells, the viral genome replication rate and protein synthesis were significantly increased in addition to glycolysis. While the glucose analogue 2-DG had no effect on RuV infection of IFNAR/IFNLR1 KO A549 cells, its application increased RuV infection on control A549 cells. Thus, glycolytic activity in the presence of RuV was influenced by IFN signaling in a context-dependent manner. It appears that glycolysis supports antiviral activity of type I IFNs. The interaction between IFN signaling pathways and glycolysis contributes to our understanding of antiviral mechanisms on epithelial cells as part of the barrier structures against virus infections.

## 2. Results

### 2.1. Enhancement of RuV Infection in the Absence of Type I and III IFN Signaling

Infection of control A549 and A549 cells with a loss-of-function deletion of the type I IFN receptor IFNAR either solely (IFNAR KO) or in combination with IFNLR1 (IFNAR/IFNLR1 KO) allowed for the analysis of the antiviral efficacy of endogenous type I and III IFNs that are generated during RuV infection, as shown in our own publication [10]. First, we compared genome replication between these three cell types at an early (24 hours post-infection [hpi]) and late (96 hpi) time point during infection. Although replication rate was already at 24 hpi slightly higher in IFNAR/IFNLR1 KO, significant differences to the control cells were detected at 96 hpi (Figure 1A). Already in the absence of type I IFN signaling, a significant increase over the control cells was detected, which was even higher in IFNAR/IFNLR1 KO. The antiviral efficacy of not only type I, but also type III IFN signaling was also present at protein level. Western blot analysis of RIPA lysates prepared at 24 and 96 hpi (Figure 1B) revealed a tendency similar to the number of viral genome copies (Figure 1A), which was an increase starting at 24 hpi on IFNAR/IFNLR1 KO A549 until a notable increase over the control cells was reached at 96 hpi (Figure 1B). Quantification of E1 expression level by densitometric analysis confirmed these differences between the three cell types, which were significant for IFNAR/IFNLR1 KO at 96 hpi in comparison to control A549 at 96 hpi (Figure 1C). As a reference for the absence of IFN signaling after receptor KO, Figure 1C also shows the expression level of phosphorylated and thus activated STAT1 as the central signaling component of the type I and III IFN pathways. Besides RuV infection, exogenous IFN β (10 ng/mL) was included as a positive control. As expected, exogenous IFN β induced phosphorylation of STAT1 (p-STAT1) in control A549 cells, but not in IFNAR KO and IFNAR/IFNLR1 KO cells. The p-STAT1 signal increased from 1 to 4 dpi in RuV-infected A549 control cells and was, in comparison to the control A549 cells, present at a reduced intensity in IFNAR KO A549 (Figure 1C). This is probably based on type III IFN signaling, as in the presence of RuV, no p-STAT1 signal was detected in IFNAR/IFNLR1 KO A549 cells (Figure 1C). Hereafter, we addressed the distribution of RuV-positive cells at 96 hpi by immunofluorescence analysis (Figure 1F). This qualitative analysis revealed a comparable distribution of infected cells within the monolayer of the three A549 cell types. 

In conclusion, although RuV infection was enhanced in the absence of type I IFNs, type III IFNs appear to have, thus far, an underrated contributory role in the cellular antiviral response of epithelial cells to RuV.

### 2.2. Type III IFN Signaling Influences the Synthesis of IFN β in the Absence of Type I IFN Signaling

After examining the impact of type I and III IFNs on the establishment of RuV infection in IFN receptor KO cells, we next determined the influence of IFN signaling on IFN production itself. A549, carrying IFNAR KO alone or in combination with the IFNLR1 KO (IFNAR/IFNLR1 KO), were infected with RuV followed by determination of the IFN (α2, β, γ, λ1, and λ2/3) level in the supernatant. At 24 hpi, no IFN was detected on RuV-infected A549 cells (data not shown). Figure 2 indicates that at 96 hpi A549 carrying IFNAR KO produced significant higher amounts of IFN β compared to control A549. However, on IFNAR/IFNLR1 KO A549, no IFN β was detected, while type III IFNs were still produced, albeit at a significantly reduced level. All three cell types lacked IFN α2 and IFN γ generation in response to RuV infection. In conclusion, IFN signaling appears to influence the generation of IFNs during RuV infection. In the absence of type I IFN signaling, the generation of type I IFNs during RuV infection was significantly increased, while in the absence of both type I and III IFN signaling, the production of IFN β as well as λ was significantly reduced during RuV infection.

### 2.3. In the Absence of Type I and III IFN Signaling Glycolysis Was Increased during RuV Infection

Next, metabolic activity in RuV-infected A549 cells in the absence of IFN signaling was analyzed by extracellular flux analysis with the cell energy phenotype test kit. For the rate of mitochondrial respiration (oxidative phosphorylation), the oxygen consumption rate (OCR) was measured and for glycolysis, the extracellular acidification rate (ECAR). However, it needs to be considered that lactate can be oxidized at mitochondria and that other metabolic pathways such as the TCA cycle can also contribute to extracellular acidification [16]. The increase in OCR after trifluoromethoxy carbonylcyanide phenylhydrazone (FCCP) injection is based on depolarization of the mitochondrial membrane and loss in the mitochondrial membrane potential as the cells try to compensate for this loss through an increased oxygen uptake. The inhibition of the ATP synthase with oligomycin results in the loss of the energy supply by mitochondrial respiration, which in turn leads to an increase in glycolysis as a compensatory mechanism. Figure 3A,C show the graphical representation of eight measurement points for OCR and ECAR, respectively, over a total time period of 50 min. Irrespective of the presence or absence of IFN-associated signaling in A549 cells, Figure 3A indicates a shift in OCR in all three A549 cells to higher values in the presence of RuV. This was confirmed by the calculation of basal and stressed (after injection of the inhibitors of mitochondrial oxidative phosphorylation) OCR (Figure 3B): For all three A549 cell types addressed with our extracellular flux measurements, a significant increase in stressed OCR was detected after infection with RuV. Figure 3C shows a graphical representation of the ECAR points during extracellular flux measurement, which indicates an increase in basal glycolysis after RuV infection. The calculation of stressed OCR and ECAR revealed differences among the three addressed cell types (Figure 3D): basal ECAR was significantly increased in IFNAR/IFNLR1 KO A549 after RuV infection, whereas ECAR was similar to the mock in the control and IFNAR KO A549 cells. Moreover, the highest increase in stressed ECAR was observed for A549 IFNAR/IFNLR1 KO. The calculation of the metabolic potential, which refers to the ability of a cell to respond to an increased energy demand as posed here by the addition of inhibitors of mitochondrial oxidative phosphorylation, highlights similarities between mock and RuV infection. The metabolic potential of mitochondrial respiration was comparable between mock and RuV infection. Although the stressed OCR values were significantly higher during RuV infection, the increase in basal OCR appeared to result in a metabolic potential similar to the mock controls. These observations do not appear to be influenced by the absence of IFN signaling on the KO cells. The same applies to the metabolic potential of glycolysis in A549 IFNAR/IFNLR1 KO cells. In conclusion, IFN-associated signaling does not appear to influence the impact of RuV on mitochondrial respiration, as mitochondrial respiration during RuV infection of the control and IFNAR/IFNL1R KO A549 cells was comparable. However, in the absence of IFN signaling, especially of type I IFN signaling, glycolysis was significantly increased during RuV infection.

### 2.4. Type I and III IFN Signaling Influence the Requirement of RuV for Glycolysis

We next extended our observations on the increase in glycolysis in IFNAR/IFNLR1 KO cells during RuV infection to the characterization of the contribution of glycolysis to the course of RuV infection. The glucose analogue 2-DG acts on hexokinase II and thus inhibits glycolysis. 2-DG reduced RuV infection on Vero cells [11], which due to a deletion within the type I IFN genes do not produce type I IFNs [12]. Here, we extended these observations on the contribution of glycolysis to RuV infection to A549 cells lacking either just type I IFN signaling or both type I and III IFN signaling. The addition of 2-DG was performed at 24 and 72 hpi to address the early and late effects during RuV infection. As a first indication, qualitative analysis by immunofluorescence showed that in the absence of type I and III IFN signaling, 2-DG had no effect on RuV-positive cells (Figure 4A). The observed semiquantitative increase in control A549 after the application of 2-DG at 24 hpi was also confirmed by quantitative Western blot analysis (Figure 4B). Densitometric analysis of the Western blot results confirmed that the application of 2-DG, especially at 24 hpi, but also 72 hpi, led to a significant increase in E1 protein expression in the control A549 cells. There was a tendency for a reduced expression in IFNAR KO A549 cells, which was not significant. However, there was no effect detected for IFNAR/IFNLR1 A549 (Figure 4C). In conclusion, the contribution of glycolysis to RuV infection was influenced by the presence of type I and III IFN signaling.

Figure 5 sets the results on the IFNAR/IFNLR1 KO A549 cells in the context of the interaction between IFN generation and IFN signaling and glycolysis. As described by Ren and colleagues, glycolysis and the antiviral IFN pathway are connected, which includes the association of mitochondrial antiviral-signaling protein (MAVS) with hexokinase II (HKII) as the key glycolytic enzyme catalyzing the reaction of glucose (Glc) to glucose-6-phosphate (G6P) [17]. As summarized in the review by Ren et al, MAVS localizes to the outer membrane of mitochondria and serves as an adaptor downstream of the dsRNA recognition receptor RIG-I (retinoic acid inducible gene I) and is thus required for the generation of IFNs. The switch of MAVS binding from HKII to RIG-I reduces glycolysis. In the absence of IFN signaling on IFNAR/IFNLR1 KO A549, the synthesis of RIG-I as an IFN-stimulated gene is not further induced and thus MAVS could be available for HKII, which could support the observed increase in glycolysis. However, glycolysis also contributes to the antiviral activity of IFNs as under certain conditions, glycolysis is positively involved in the activation of STAT1 and STAT2 and subsequently in the expression of ISGs [18]. Additionally, the activation of protein translation by IFN signaling is a process that requires energy that in turn is at least partially provided by glycolysis. This could explain the higher replication rate after application of 2-DG to control A549 cells as this would reduce the effect of antiviral IFN response. Moreover, glycolysis could also at least partially support virus replication as appears to be the case for RuV infection in the absence of type I IFN signaling, while type III IFN signaling is maintained.

## 3. Discussion

The antiviral action of IFNs is complex and involves hundreds of ISGs, which are expressed in a cell type-specific manner. These ISGs do not only interfere with various aspects of the viral replication cycle including genome replication and viral protein translation as addressed here, but also influence metabolic activity and as such, a source for cytokine synthesis, but also for viral energy supply. Here, we addressed not only the contribution of IFNs to metabolic alterations as an important aspect of RuV infection and the possibility of rubella pathogenesis, but also measured feedback interaction between type I and III IFN-associated signaling and IFN synthesis. The IFN concentrations measured under IFNAR KO indicate that the absence of receptor-mediated signals from type I IFNs via the IFNAR had little effect on the formation of type III IFNs. The observation that the formation of IFN β under IFNAR KO even increased compared to the control cells with functional IFNAR suggests either a negative feedback from IFN β based on its own formation, or rather, from the factors that are induced through IFN β signaling or a positive circuit from type III IFN signaling. Transcriptional induction of a variety of IFN-stimulated genes such as the sensory pathway components melanoma differentiation-associated protein 5 (MDA5) and RIG-I or the transcription factor interferon regulatory factor 7 (IRF7) enhance the formation of type I and type III IFNs [19]. To maintain tissue homeostasis and avoid adverse effects from the potent pro-inflammatory effects of type I IFNs, the strength and duration of the IFN response must be tightly regulated [20,21]. One inducible negative feedback regulator of type I IFN signaling is SOCS1, which transiently downregulates STAT phosphorylation in the first hours [19]. Another negative regulator is ubiquitin-specific peptidase 18 (USP18), which is induced at a later time point but remains highly expressed for days [22]. Blumer et al. used KO cell lines and KO mice to demonstrate that IFN λ signaling is regulated by SOCS1, but not by SOCS3 or USP18 [19]. This mode of signaling regulation explains the differences in the kinetic properties of type I and type II IFNs. Thus, IFN λ is more prone to long-lasting IFN-dependent gene induction [19].

The A549 KO cells described here for RuV were also characterized for USUV infection. Similar to our observations here for RuV, infection of A549 cells with KO of the receptor for type I and III IFNs resulted in an increase in glycolysis compared to the control cells [14]. However, in contrast to RuV, USUV infection led only to a significant increase in basal glycolysis besides a significant reduction in the metabolic potential of glycolysis under stressed conditions. The impact of both USUV and RuV on mitochondrial respiration appears to be independent from IFN signaling.

Based on our observations on the compensatory role of type III IFNs on alveolar epithelial A549 cells in the absence of type I IFN signaling, the question arises as to why polymorphisms in IFNAR2 have such a severe consequence with an altered pathogenesis of the RuV vaccine strain. A patient with a polymorphism in IFNAR2 developed fatal encephalitis with the presence of RuV vaccine strain in a brain autopsy [8]. Based on the additional presence of mumps virus and HHV-6, the contribution of RuV to the pathology in the CNS was not possible to resolve, but this is an unusual finding for the RuV vaccine strain. The antiviral efficacy of IFNs is cell type-specific. A549 cells are epithelial cells from the respiratory tract and the patient with the polymorphism in IFNAR2 did not suffer from severe diseases caused by respiratory viruses that cause common childhood infections prior to MMR vaccination [8]. The relevance of both type I and III IFNs for the antiproliferative action of the IFN system during virus infections could be a property of respiratory cells such as A549, as type III IFNs are especially active at mucosal surfaces in the respiratory tract [23]. However, RuV, in contrast to respiratory tract viruses, also causes a systemic infection. At the blood–brain barrier (BBB), for example, type I IFN β has a distinctive role through its rather stabilizing effect on the permeability of the BBB after administration of histamine [24]

There is a complex and often cell type-specific interdependency of cellular metabolism and antiviral IFN response. The immune response itself including the synthesis of cytokines requires energy and metabolic intermediates. On plasmacytoid dendritic cells (pDCs), influenza virus infection induced an IFN-independent upregulation of glycolysis, but glycolysis was required for an antiviral response and the activation state of pDCs [25]. As such, Bajwa and colleagues reported a glycolysis-associated increase in IFN generation. Notably, on H1N1-infected A549 cells, the treatment with the 2-DG and oxamate as inhibitors of glycolysis led to a decrease in the transcription of IFN α and selected ISGs. Moreover, 2-DG was discussed as a therapeutic strategy against SARS-CoV-2 as it inhibited its replication in Caco-2 cells as a colon carcinoma cell line [26] and the thus discussed requirement of SARS-CoV-2 replication for glucose metabolism [27]. However, SARS-CoV-2 infection is also associated with a decreased production of type I IFN despite an enhanced production of pro-inflammatory cytokines [28,29]. The decrease in type I IFNs is mediated by the nucleoprotein N, which prevents the TRIM25-mediated activation of the RIG-I pathway [28]. Our data on an improved RuV infectious cycle after application of 2-DG to control A549 are also supported by the observation of a regulation circuit between IFN β and glycolysis [30]. Glycolysis is required for the rapid induction of an antiviral response against coxsackievirus B3 as inhibition of glycolysis with 2-DG reduced the IFN-associated antiviral response [30]. These observations are consistent with our results on an increase in viral E1 protein synthesis on A549 control cells after the application of 2-DG. In the absence of type I IFN, but maintained type III IFN signaling, 2-DG acts to some degree as an antiviral, whereas in the presence of both type I and III IFN signaling, 2-DG acts as a proviral. Thus, in an IFN signaling-dependent manner, cellular countermeasures against RuV appear to be supported by glycolysis. Future studies are required to elucidate the impact of 2-DG on the activation of selected ISGs in the presence and absence of type I IFN signaling. The differential impact of the virus infection itself and the infection-associated IFN response need to be considered in the characterization of cellular metabolism during virus infections and thus in the consideration of targets within glycolysis as an antiviral treatment option.

## 4. Materials and Methods

### 4.1. Reagents

The glucose analogue 2-DG (Santa Cruz Biotechnologies, Heidelberg, Germany) was dissolved in PBS at a final concentration of 50 mM and stored in aliquots at −20 °C. 2-DG was added to the cultivation medium to achieve a final concentration of 5 mM. IFN β was purchased from Peprotech (Hamburg, Germany) and was dissolved according to the manufacturer’s instructions and stored in aliquots at −20 °C. Fluoromount G and reagents for cell cultivation were from Thermo Fisher Scientific (Carlsbad, CA, USA).

### 4.2. RNA Extraction and Quantification of Viral Genome Copies

Taq Man-based one-step reverse transcription quantitative PCR (TaqMan RT-qPCR) was used as described with minor modifications [31]. Briefly, total RNA was extracted by the ReliaPrep™ RNA Miniprep Systems (Promega, WI, USA). For amplification of a conserved region within the viral p90 gene, the sense primer RV_235.s, 5’-CTG CAC GAG ATY CAG GCC AAA CT-3’, the antisense primer RV_419.as, 5’-ACG CAG ATC ACC TCC GCG GT-3’, and the TaqMan fluorogenic probe RV_291TaqFAM, 6FAM-TCA AGA ACG CCG CCA CCT ACG AGC-BBQ were employed together with the Qiagen (Hilden, Germany) QuantiTect Probe RT-PCR Kit.

### 4.3. Cultivation of Permanent Cell Lines

The adenocarcinomic alveolar basal epithelial A549 control and KO cell lines were generated through CRISPR-Cas9 technology as described [32]. Briefly, transduction with lentiviral vectors (with kind provision of the lentiCRISPR v2 system by Feng Zhang) was used for the generation of A549 KO cell lines by CRISPR/Cas9 technology. Guide RNAs were ordered as dsDNA oligos. For cloning into the lentiCRISPR v2 plasmid (#52961, Addgene, Watertown MA, USA), the BsmBI site was used, which was followed by the generation of lentiviral particles as previously described [32]. Single cell clones were obtained through limiting dilution. For their cultivation, the following maintenance medium was used (all components were from ThermoFisher Scientific, Carlsbad, CA, USA): Dulbecco’s modified Eagle’s medium (DMEM) with GlutaMAX and high-glucose under supplementation with 10% fetal calf serum (FCS), 1% non-essential amino acids (MEM NEAA), and 100 U/mL penicillin and 100 mg/mL streptomycin (Pen/Strep). For RuV titration by the plaque assay, Vero cells (ATCC CCL-81) were used and cultivated accordingly, but without the addition of NEAAs. All cell lines were passaged with 0.05% trypsin and were cultivated at 37 °C in a humidified atmosphere.

### 4.4. RuV Strains, Titer Determination, and Virus Infection

For infection with RuV, the clinical isolate RVi/Wuerzburg.DEU/47.11_12-00009 (Wb-12, genotype 2B) was used (up to passage 10 after preparation on Vero cells). Titer of stock virus was determined on Vero cells by the plaque assay with agar overlay and followed by staining with crystal violet solution. Infection was conducted at an MOI of 5 using DMEM supplemented with 1% FCS and an incubation period of two hours at 37 °C. Medium change comprised washing with PBS followed by the addition of maintenance medium.

### 4.5. Measurement of IFN in Culture Supernatants

IFN concentration in culture supernatants of the A549 control and KO cells (seeded at a density of 1 × 10^5^ cells per well of a 24 well plate in 0.5 mL) were measured at 96 hpi using the LEGENDPLEX human type 1/2/3 Interferon panel (5-plex) kit (BioLegend, San Diego, CA, USA) according to the manufacturer’s protocol.

### 4.6. Metabolic Flux Measurement and Analysis of the Metabolic Phenotype and the Metabolic Potential 

Metabolic activity in the A549 control and KO cells was determined as real-time measurement with the XFp extracellular flux analyzer (Agilent Seahorse Bioscience Technologies, Santa Clara, CA, USA). In reference to mitochondrial respiration (oxidative phosphorylation) and glycolysis, OCR and ECAR were determined, respectively [33]. Measurements were based on the cell energy phenotype test kit (Agilent Seahorse Bioscience Technologies, Santa Clara, CA, USA) based on co-injection of the ATP-synthase inhibitor oligomycin (at 1 µM) and FCCP (at 0.8 µM). All metabolic measurements were conducted in duplicates per sample (n = 2, experimental replicates) and repeated three times (n = 3, biological replicates). Plating of cells was conducted at 1.6 × 10^4^ cells per well of the XFp miniplates. Prior to metabolic flux measurement, the cultivation medium was changed to XF DMEM (Agilent Seahorse Bioscience Technologies) supplemented with 10 mM glucose (Agilent Seahorse Bioscience Technologies), 1 mM sodium pyruvate (Sigma Aldrich, Taufkirchen, Germany), and 2 mM L-glutamine (ThermoFisher Scientific, Carlsbad, CA, USA) followed by incubation for 1 hour at 37 °C in a CO_2_-free incubator. Measurements were followed by the assessment of the metabolic phenotype and the metabolic potential by the Seahors XF cell energy phenotype test report generators. Metabolic potential is based on the increase in stressed (after injection of the inhibitors) OCR and ECAR over basal (baseline) OCR and ECAR, respectively. It describes the capacity of a given cell to respond to a rise in the cellular energy demand induced by the inhibitors of mitochondrial oxidative phosphorylation (here oligomycin and FCCP). Normalization was based on protein content, which was determined by the Bradford assay (RotiRQuant, Roth, Germany). OCR and ECAR values are shown as (pmol/min)/optical density (OD) and (mpH/min)/OD, respectively.

### 4.7. Immunofluorescence Analysis

For immunofluorescence-based detection of the RuV antigen, A549 cells were cultivated on glass slides, washed at 96 hpi with PBS, and fixed with 2% (*w*/*v*) paraformaldehyde (PFA) in PBS. An additional wash step with PBS was followed by permeabilization with 0.3% Triton X-100 (*v*/*v*) in PBS for 30 min at 37 °C followed by blocking with 0.3% Triton X-100 (*v*/*v*) and 5% normal goat serum (*w*/*v*) in PBS for 30 min at 37 °C in a humidified chamber. Thereafter cells were incubated with primary anti-C antibody (clone 2-36, 1:200 dilution, Meridian Life Science, TN, USA) for 60 min at 37 °C and with secondary Cy3 conjugated donkey anti-mouse IgG secondary antibodies (1:200 dilution; Dianova, Hamburg, Germany) for 45 min followed by three washing steps with PBS. DNA counterstaining occurred during the mounting step with Flouromount G containing DAPI. Stained cell samples were analyzed with the Olympus XM10 fluorescence microscope.

### 4.8. Western Blot Analysis

After a washing step with cold PBS cell lysis was carried out with RIPA buffer (50 mM Tris, 150 mM NaCl, 1% Nonidet P-40, 0.5% deoxycholate, 0.1% SDS; pH 7.5) supplemented with complete EDTA-free protease inhibitor cocktail (Roche Diagnostics, Mannheim, Germany) and with phosphatase inhibitors (1 mM Na_3_VO_4_ and 50 mM NaF). Cell lysate was then sonicated on ice, which was followed by centrifugation at 15,000 g at 4 °C for 5 min. Supernatants were subjected to protein concentration determination by the DC Protein Assay (Bio-Rad, Hercules, CA, USA) or were stored after boiling at 95 °C for 5 min in equal volumes with 1× Laemmli sample buffer. Samples were run for SDS-PAGE on a 8% (*v*/*v*) stacking and a 12% (*v*/*v*) running SDS-polyacrylamide gel (Protean II, Bio-Rad GmbH) at a total concentration of 25 to 30 µg. Samples were transferred to a polyvinylidene difluoride (PVDF) membrane (Amersham Biosciences, Munich, Germany). PVDF membranes were blocked with 5% milk powder and probed with the following antibodies (Ab) at the indicated dilutions: anti-phospho-STAT1 rabbit Ab (Tyr701, D4A7, 1:1000); anti-rubella E1 mouse Ab (1:500, Merck, Darmstadt, Germany) and anti-β-actin mouse Ab (clone AC 74, 1:2000, Sigma-Aldrich, St. Louis, MO, USA). Primary antibodies were detected with the following POD-conjugated secondary antibodies: goat anti-rabbit IgG Ab (1:20,000, Dianova, Hamburg, Germany) or goat anti-mouse IgG Ab (1:8000, Sigma-Aldrich, St. Louis, USA). To visualize proteins, chemiluminescent detection on PVDF membranes was carried out with ECL-A/ECL-B substrate (both from Sigma Aldrich, St. Louis, MO, USA) or SuperSignal West Femto Maximum Sensitivity Substrate (Thermo Fisher Scientific, Carlsbad, CA, USA) and analyzed on a Luminescent Image Analyzer (LAS 1000, Fujifilm, Tokyo, Japan). Obtained protein bands were subjected to densitometric analysis with the AIDA Image Analyzer software 4.1 (Elysia-raytest GmbH, Straubenhardt, Germany) and p-STAT1 and E1 protein expression were expressed as density relative to β-actin.

### 4.9. Statistical Analysis

Data were analyzed by GraphPad Prism software 9.1.2 (GraphPad Software, San Diego, CA, USA) by unpaired t-test and one-way analysis of variance (ANOVA) and Tukey’s post-hoc multiple comparisons test. Data are shown as the mean ± standard deviation (SD). Statistical significance is indicated as * *p* < 0.05, ** *p* < 0.01, *** *p* < 0.001, and **** *p* < 0.0001.

## Figures and Tables

**Figure 1 pathogens-11-00537-f001:**
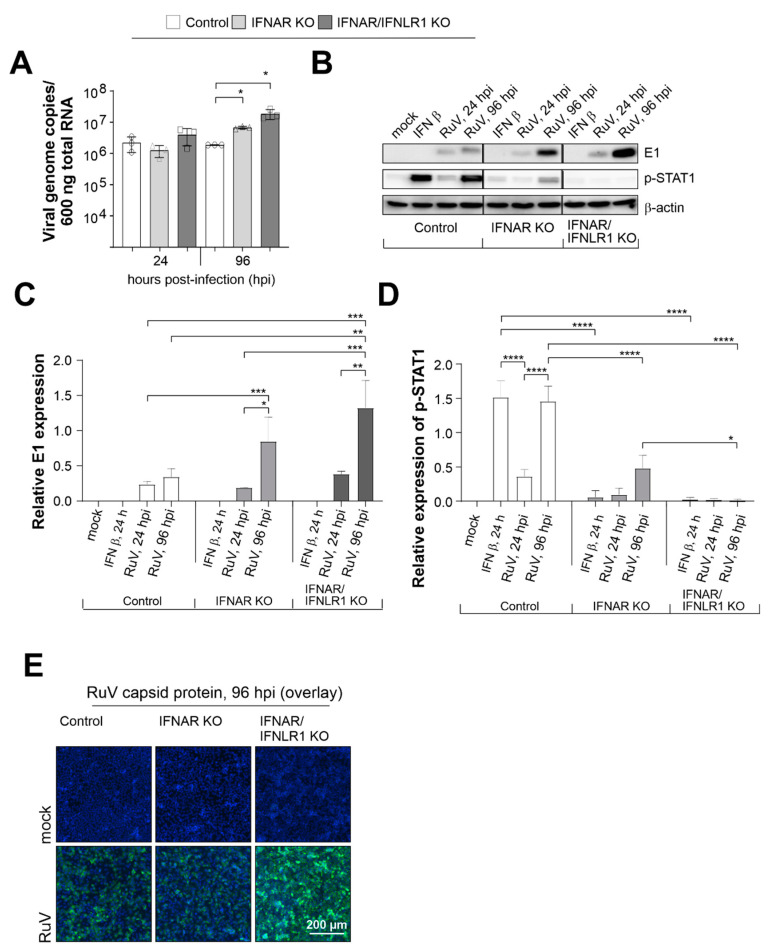
In the absence of IFN signaling, RuV genome replication and protein synthesis are increased. A549 cells with KO of the type I IFN receptor IFNAR either solely or together with the type III IFN receptor (IFNLR1) and A549 control cells (1 × 10^5^ cells) were infected with RuV (multiplicity of infection [MOI] of 5). (**A**) At the indicated time points, viral genome copies per 600 ng of total RNA were determined by one step Taq Man PCR. Single values are indicated as circles (control A549), triangles (IFNAR KO), and squares (IFNAR/IFNLR1 KO). (**B**) Western blot analysis of RIPA lysates (25 µg protein) obtained for samples after RuV infection for indicated time points or treatment with IFN β (10 ng/mL) for 24 hours. Western blot results were subjected to densitometric analysis of (**C**) E1 and (**D**) p-STAT1 expression. (**C**,**D**) are based on arbitrary units and normalization to the housekeeping protein β-actin. Results are shown as means ± SD of n = 3 experiments. Statistical significance was determined by analysis of one-way ANOVA with Tukey’s multiple comparisons test. (**E**) Immunofluorescence analysis with antibodies against capsid protein. * *p* < 0.05, ** *p* < 0.01, *** *p* < 0.001, and **** *p* < 0.0001.

**Figure 2 pathogens-11-00537-f002:**
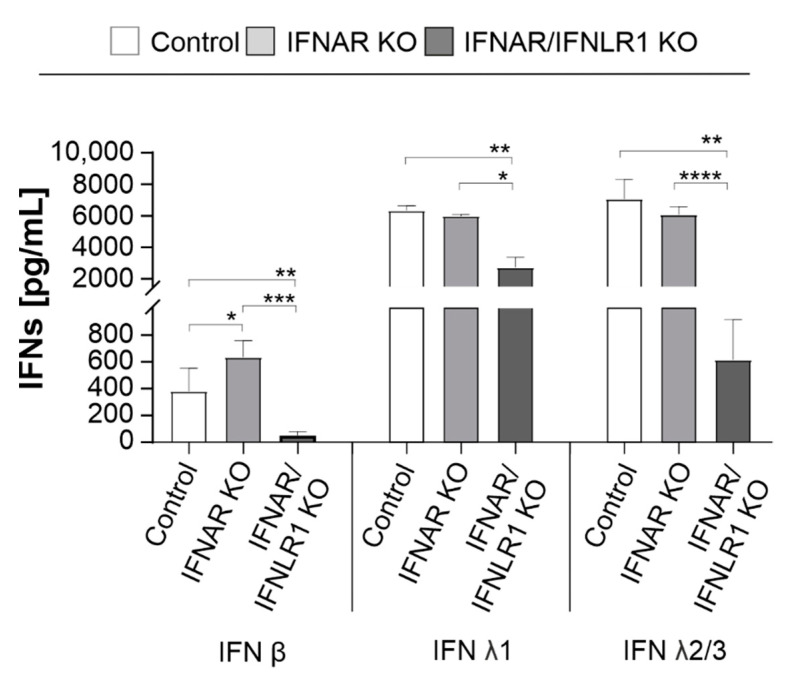
Effect of IFN signaling on the generation of IFNs during RuV infection. A549 cells with KO of the type I IFN receptor IFNAR either solely or together with the type III IFN receptor (IFNLR1) and A549 control cells (1 × 10^5^ cells) were infected with RuV (MOI of 5). IFN protein levels in the supernatant were determined at 96 hpi by the LEGENDPLEX human type 1/2/3 Interferon panel (5-plex) kit. Data are shown as means ± SD (n = 3). Statistical assessment for each IFN was performed by one-way ANOVA and Tukey’s post-hoc multiple comparisons test. * *p* < 0.05, ** *p* < 0.01, *** *p* < 0.001, and **** *p* < 0.0001.

**Figure 3 pathogens-11-00537-f003:**
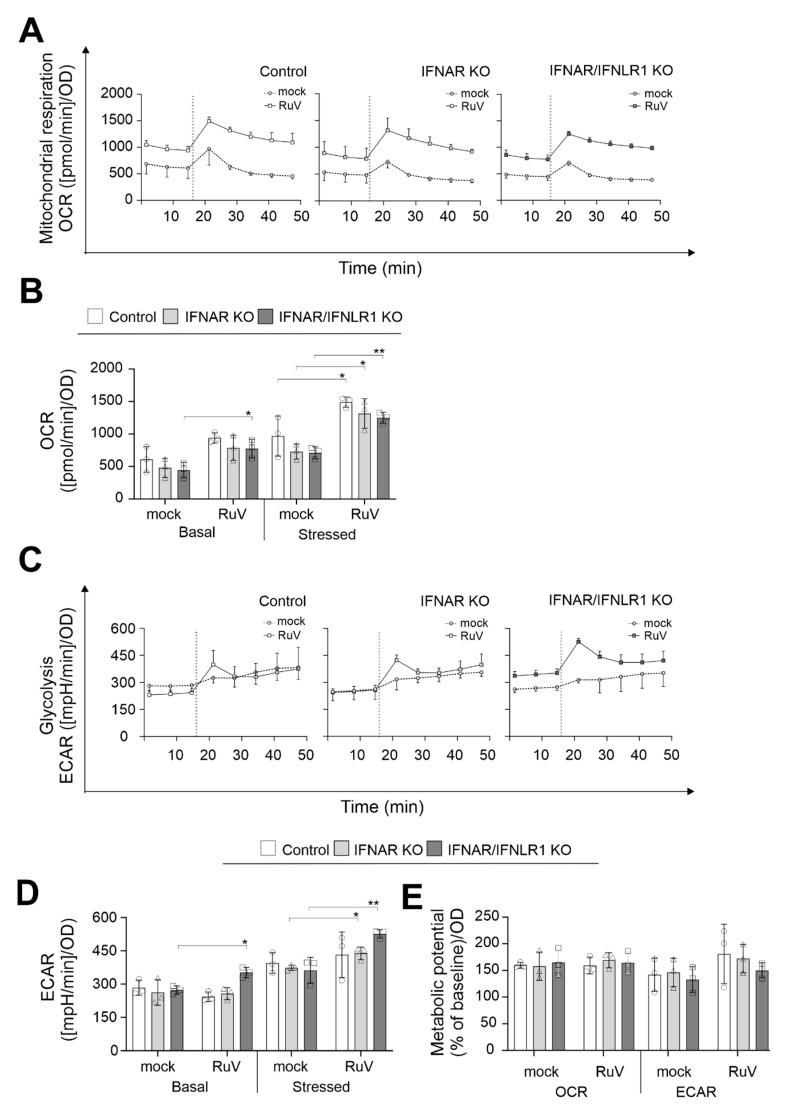
In the absence of type I and III IFN signaling, basal and stressed glycolysis were increased during RuV infection, whereas mitochondrial respiration was not affected. (**A**,**C**) Graphical representation of the extracellular flux measurement with the cell energy phenotype test kit performed at 96 hpi for the indicated A549 cell types infected with an MOI of 5. Injection of 1 µM oligomycin and 0.8 µM FCCP is indicated by a dashed line. Data for (**A**) OCR and (**C**) ECAR are shown as mean ± SD (n = 3). (**B**,**D**) Quantification (means ± SD, n = 3) of (**B**) OCR and (**D**) ECAR under basal and stressed (as induced by inhibitors of mitochondrial oxidative phosphorylation) conditions. (**E**) Calculation of the metabolic potential based on the percent increase in stressed OCR and ECAR over the corresponding basal measurement points. (**D**,**E**) Single values are indicated as circles (control A549), triangles (IFNAR KO), and squares (IFNAR/IFNLR1 KO). (**B**,**D**,**E**) Calculations are based on Seahors XF cell energy phenotype test report generators (www.agilent.com; accessed on 21 March 2022) followed by statistical assessment by the unpaired t-test in reference to the uninfected control (mock). * *p* < 0.05, ** *p* < 0.01.

**Figure 4 pathogens-11-00537-f004:**
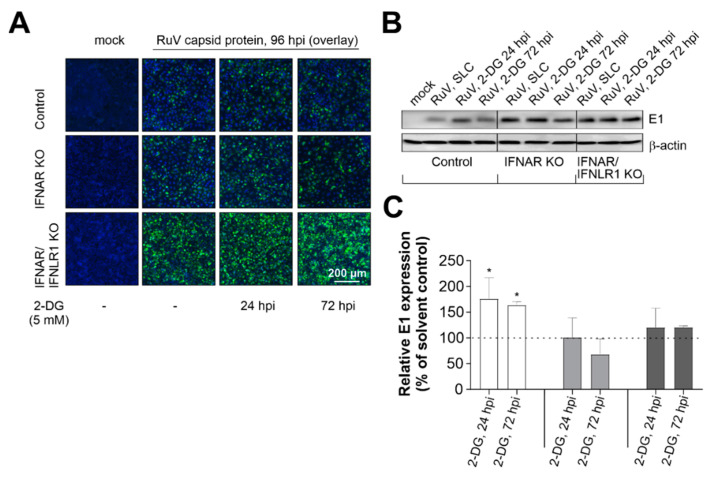
In the absence of type I and III IFN signaling, the glycolysis inhibitor 2-DG has no effect on RuV infection. (**A**) Immunofluorescence analysis at 96 hpi with anti-C antibody after application of 2-DG at 24 and 72 hpi. (**B**) Western blot analysis of RIPA lysates (25 µg protein) and densitometric analysis of (**C**) E1 expression. (**C**) E1 expression was determined as percent of solvent control based on arbitrary units and normalization to the housekeeping protein β-actin. Results are shown as means ± SD of n = 3 experiments. Statistical significance was determined by analysis of one-way ANOVA with Tukey’s multiple comparisons test in reference to the solvent control (SLC). * *p* < 0.05.

**Figure 5 pathogens-11-00537-f005:**
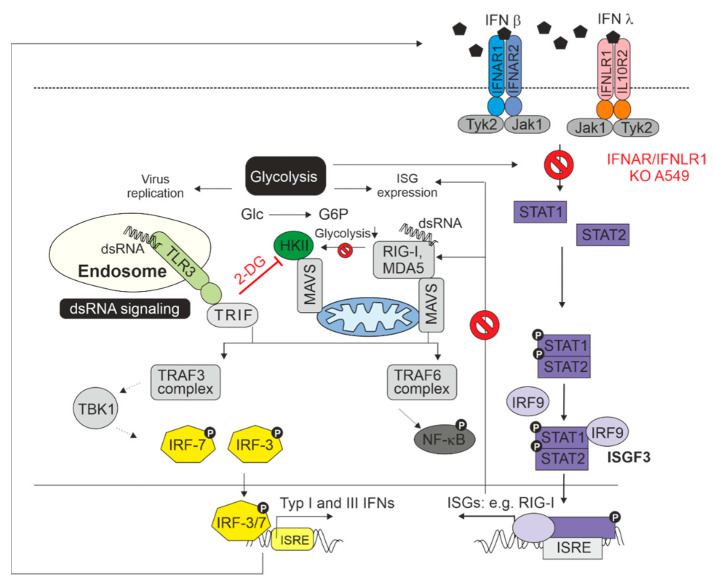
Summarizing figure. The interaction between the dsRNA sensing pathways, IFN signaling, and glycolysis in the context of a virus infection as shown here for RuV and the results highlighted in this manuscript. For more detailed information, please refer to the main text.

## Data Availability

The raw data supporting the conclusions of this article will be made available by the corresponding author without undue reservations.

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
