# Peer review of "Interferon Signaling-Dependent Contribution of Glycolysis to Rubella Virus Infection"

_pathogens, 2022, doi:10.3390/pathogens11050537_

Round 1

Reviewer 1 Report

In this study, the authors proved that the metabolic alterations induced by rubella on the human lung epithelial A549 cells are influenced by the associated IFN response. The authors reported that in the absence of both type I and III IFN signaling rubella virus replication cycle was enhanced. The interaction between glycolysis and antiviral IFN signaling in rubella virus infection was emphasized in this study.

Although lengthy, overall, the manuscript and the study results were well written and presented. More concise and well-structured sentences and paragraphs are highly recommended.

Author Response

The authors are grateful to the reviewers for their helpful comments. In addition to the reviewers’ suggestions we have also revised summarizing Figure 5, as in the original Figure 5 there was a double statement of “ISG expression”, which was revised to “Virus replication” and “ISG expression”.

Although lengthy, overall, the manuscript and the study results were well written and presented. More concise and well-structured sentences and paragraphs are highly recommended.

#Thank you very much for your kind comment. We revised and shortened the introduction, the results, and the discussion section.

Reviewer 2 Report

Comments to the manuscript by Schlling et al.:

Overall, the study was well designed and the data presented are almost sound to lead the main conclusion that glycolysis is increased during RuV infection in the absence of type I and III IFN signaling. However, I found several flaws as listed below. I consider the manuscript should be accepted after making appropriate improvements.

Major comments:

1) L128-129: The authors described, “in the absence of IFN signaling, protein expression was rather inconsistently distributed within the cell monolayer into high and low RuV capsid (C) protein-expressing cells.” I was not convinced by this argument because quantitative image data are unfortunately lacking. The authors should add the quantitative data for important imaging results: for example, FACS analysis data of the cells shown in Fig. 1E.

2) L151-152 and L159-160: The authors stated “we next determined the influence of IFN signaling on IFN production itself” and concluded,” in conclusion, in the absence of type I IFN signalling, a positive influence of type III IFNs on production of IFN β was present during RuV infection.” I could not understand the logic of this conclusion. The authors should more explain why the results of Fig. 2 lead to this conclusion. If the authors want to show just that a positive influence of type III IFNs on the production of IFN β is present, I think that the author should have examined whether adding a recombinant type III INF to the cell culture results in an enhanced IFNβproduction in IFNAR KO cells in the absence of RuV infection. Or, did the authors want to examine the influence of IFN signaling on the production of different IFN types in RuV-infected cells?

3) L174: Because ECAR is the method to simply measure the extracellular pH, this method does not necessarily determine the rate of glycolysis specifically. For example, a previous study suggested a glycolysis-independent acidification pathway as an oxamate-insensitive pathway (Am J Physiol Cell Physiol 292: C125–C136, 2007). The authors should briefly explain why ECAR is appropriate to specifically determine the rate of glycolysis under the experimental conditions used or should describe the limitation of the interpretation of the ESCAR data presented. The abbreviation ECAR should be defined at the first appearance (L174).

Minor comments:

4) L68: RuV is the sole member of the genus Rubivirus. Please briefly explain why the authors employed RuV, not more widely analyzed viruses (e.g., influenza virus), as a representative virus in this study although the study attempted to answer the basic question of whether virus infections alter the glycolysis in the host cells. Or, when the authors are specifically interested in RuV, previous studies showing that infection-dependent alterations in the glycolytic pathway by other viruses should better be acknowledged in the introduction section if any.

5) L70-91: The author can greatly shorten the last paragraph of the introduction because this part is largely redundant to the Results.

6) L184 and others: I think the term metabolic stressors is inappropriate to represent oligomycin/FCCP, which have been historically referred to as “oxidative phosphorylation inhibitors” or “energy poisons”.

7) Fig. 3A, C: The size of the symbols representing mock and RuV in the graphs is too small to distinguish them. The size should be larger (and/or use cloze/open symbols) to be easily recognized.

Author Response

Overall, the study was well designed and the data presented are almost sound to lead the main conclusion that glycolysis is increased during RuV infection in the absence of type I and III IFN signaling. However, I found several flaws as listed below. I consider the manuscript should be accepted after making appropriate improvements.

Major comments:

1) L128-129: The authors described, “in the absence of IFN signaling, protein expression was rather inconsistently distributed within the cell monolayer into high and low RuV capsid (C) protein-expressing cells.” I was not convinced by this argument because quantitative image data are unfortunately lacking. The authors should add the quantitative data for important imaging results: for example, FACS analysis data of the cells shown in Fig. 1E.

#This original statement should express, that the distribution of infected cells was similar between the three A549 cell types. As this statement was misleading, we have changed it as follows: "This qualitative analysis revealed a comparable distribution of infected cells within the monolayer of the three A549 cell types."

2) L151-152 and L159-160: The authors stated “we next determined the influence of IFN signaling on IFN production itself” and concluded,” in conclusion, in the absence of type I IFN signalling, a positive influence of type III IFNs on production of IFN β was present during RuV infection.” I could not understand the logic of this conclusion. The authors should more explain why the results of Fig. 2 lead to this conclusion. If the authors want to show just that a positive influence of type III IFNs on the production of IFN β is present, I think that the author should have examined whether adding a recombinant type III INF to the cell culture results in an enhanced IFNβproduction in IFNAR KO cells in the absence of RuV infection. Or, did the authors want to examine the influence of IFN signaling on the production of different IFN types in RuV-infected cells?

#The authors are grateful for pointing this aspect out as we intended to express the impact of IFN signalling on the generation of IFNs during RuV infection. We have revised this original statement as follows:

"In conclusion, IFN signaling appears to influence the generation of IFNs during RuV infection. In the absence of type I IFN signaling, the generation of type I IFNs during RuV infection was significantly increased, while in the absence of both, type I and III IFN signaling, the production of IFN β as well as λ was significantly reduced during RuV infection."

3) L174: Because ECAR is the method to simply measure the extracellular pH, this method does not necessarily determine the rate of glycolysis specifically. For example, a previous study suggested a glycolysis-independent acidification pathway as an oxamate-insensitive pathway (Am J Physiol Cell Physiol 292: C125–C136, 2007). The authors should briefly explain why ECAR is appropriate to specifically determine the rate of glycolysis under the experimental conditions used or should describe the limitation of the interpretation of the ESCAR data presented. The abbreviation ECAR should be defined at the first appearance (L174).

#We have added to our manuscript the following statement and reference:

"However, it needs to be taken into account that lactate can be oxidized at mitochondria and that other metabolic pathways such as the TCA cycle can also contribute to extracellular acidification."

"Schmidt, C.A.; Fisher-Wellman, K.H.; Neufer, P.D. From OCR and ECAR to energy: Perspectives on the design and interpretation of bioenergetics studies. J Biol Chem 2021, 297, 101140, doi:10.1016/j.jbc.2021.101140."

Minor comments:

4) L68: RuV is the sole member of the genus Rubivirus. Please briefly explain why the authors employed RuV, not more widely analyzed viruses (e.g., influenza virus), as a representative virus in this study although the study attempted to answer the basic question of whether virus infections alter the glycolysis in the host cells. Or, when the authors are specifically interested in RuV, previous studies showing that infection-dependent alterations in the glycolytic pathway by other viruses should better be acknowledged in the introduction section if any.

#We have revised the introduction accordingly to clarify the aspect on the relevance of our study for RuV and of the interaction between metabolic pathways and the interferon system during virus infections in general.

5) L70-91: The author can greatly shorten the last paragraph of the introduction because this part is largely redundant to the Results.

#Done as suggested.

6) L184 and others: I think the term metabolic stressors is inappropriate to represent oligomycin/FCCP, which have been historically referred to as “oxidative phosphorylation inhibitors” or “energy poisons”.

#As suggested, we have replaced the term “metabolic stressors” by “inhibitors of mitochondrial oxidative phosphorylation”.

7) Fig. 3A, C: The size of the symbols representing mock and RuV in the graphs is too small to distinguish them. The size should be larger (and/or use cloze/open symbols) to be easily recognized.

#We have revised the graphs accordingly. The mock is now characterized by a dashed lined and an open symbol, while the RV graphs have a closed symbol with the colour depicted in the bar graphs depicted in Figure 3B and 3D.

Reviewer 3 Report

Very nice study clearly reported

Author Response

The authors are grateful to the reviewers for their helpful comments. In addition to the reviewers’ suggestions we have also revised summarizing Figure 5, as in the original Figure 5 there was a double statement of “ISG expression”, which was revised to “Virus replication” and “ISG expression”.

Thank you very much for your kind comment.